# Lung UltraSound Targeted Recruitment (LUSTR): A Novel Protocol to Optimize Open Lung Ventilation in Critically Ill Neonates

**DOI:** 10.3390/children9071035

**Published:** 2022-07-12

**Authors:** Roberto Chioma, Lorenzo Amabili, Elena Ciarmoli, Roberto Copetti, Pier Giorgio Villani, Miria Natile, Giovanni Vento, Enrico Storti, Maria Pierro

**Affiliations:** 1Dipartimento Universitario Scienze della Vita e Sanità Pubblica, Unità Operativa Complessa di Neonatologia, Fondazione Policlinico Universitario A Gemelli Istituto di Ricovero e Cura a Carattere Scientifico, Università Cattolica del Sacro Cuore, 00168 Rome, Italy; roberto.chioma@gmail.com (R.C.); giov.vento@gmail.com (G.V.); 2Bernoulli Institute for Mathematics, Computer Science and Artificial Intelligence, University of Groningen, 9712 CP Groningen, The Netherlands; lorenzo.amabili@gmail.com; 3Department of Pediatrics, ASST Vimercate, Vimercate Hospital, 20871 Vimercate, Italy; elena.ciarmoli@gmail.com; 4Emergency Department, Latisana General Hospital, 33053 Udine, Italy; robcopet@gmail.com; 5Department of Critical Care, Maggiore Hospital, 26100 Cremona, Italy; pgvill@gmail.com (P.G.V.); enrico.storti@asst-cremona.it (E.S.); 6Neonatal Intensive Care Unit, Azienda Sanitaria Romagna, Infermi Hospital Rimini, 47923 Rimini, Italy; miria.natile@auslromagna.it; 7Neonatal and Paediatric Intensive Care Unit, M. Bufalini Hospital, AUSL Romagna, 47521 Cesena, Italy

**Keywords:** respiratory insufficiency, ventilator-induced lung injury, intensive care, neonatal, respiratory distress syndrome, mechanical ventilation, point-of-care diagnostics

## Abstract

This study investigated the effectiveness of an original Lung UltraSound Targeted Recruitment (LUSTR) protocol to improve the success of lung recruitment maneuvers (LRMs), which are performed as a rescue approach in critically ill neonates. All the LUSTR maneuvers, performed on infants with an oxygen saturation/fraction of inspired oxygen (S/F) ratio below 200, were included in this case−control study (LUSTR-group). The LUSTR-group was matched by the initial S/F ratio and underlying respiratory disease with a control group of lung recruitments performed following the standard oxygenation-guided procedure (Ox-group). The primary outcome was the improvement of the S/F ratio (Delta S/F) throughout the LRM. Secondary outcomes included the rate of air leaks. Each group was comprised of fourteen LRMs. As compared to the standard approach, the LUSTR protocol was associated with a higher success of the procedure in terms of Delta S/F (110 ± 47.3 vs. 64.1 ± 54.6, *p* = 0.02). This result remained significant after adjusting for confounding variables through multiple linear regressions. The incidence of pneumothorax was lower, although not reaching statistical significance, in the LUSTR-group (0 vs. 14.3%, *p* = 0.15). The LUSTR protocol may be a more effective and safer option than the oxygenation-based procedure to guide open lung ventilation in neonates, potentially improving ventilation and reducing the impact of ventilator-induced lung injury.

## 1. Introduction

Lung ultrasound (LUS) has been used in adult and pediatric patients to improve the safety and the efficacy of open lung ventilation [1,2]. Open lung ventilation represents a well-known approach to evenly distribute the tidal volume throughout the mechanically ventilated lungs, through lung recruitment maneuvers (LRMs) [3,4,5], with the aim to minimize ventilator-induced lung injury (VILI) and improve the overall outcome. In a recent randomized trial [6], elective LUS-guided recruitment, performed in premature infants suffering from respiratory distress syndrome (RDS), was proved to be more effective than the standard oxygenation-based procedure in terms of the duration of invasive ventilation, length of stay, and intra-alveolar concentration of IL-6 [6]. However, the role of ultrasound-guided LRMs as a rescue approach in critically ill infants has not been evaluated yet. The lungs of these patients, who are highly unstable, may not always be “recruitable” and the risk of failure and clinical deterioration after LRMs is increased. Therefore, it is crucial to select the cases that would benefit the most from the procedure, minimizing the risk of pulmonary and hemodynamic side effects. In a recent case series [7], our group described a novel sonographic pattern, the Sunray(S)-pattern, that develops early during rescue LRMs as a sign of lung recruitability [7,8,9]. This study aimed to compare the effectiveness of rescue LUS-guided LRMs, showing signs of lung recruitability to the traditional oxygenation-guided maneuvers in critically ill ventilated neonates.

## 2. Materials and Methods

### 2.1. Patients, Equipment, and Study Groups

All the LUS-guided recruitments, starting from a consolidation pattern (C-pattern) and showing signs of lung recruitability performed between January 2019 and August 2021 were included in this case-control study and constituted the Lung UltraSound Targeted Recruitment (LUSTR) group. The recruitable lung was determined by the development of the S-pattern during pressure augmentation; this was based on our previous report [7]. The S-pattern is characterized by the presence of vertical hyperechoic artifacts starting from the reopening bronchi, representing a sonographic sign of lung recruitability. As these linear artifacts resemble the sun rays crossing the clouds, they were named Sunray(S)-lines. The pattern is characterized by the appearance of at least one S-line per field [7]. LRMs were excluded from the study if no S-pattern was detected during the procedure.

The LRM was performed in the case of severe respiratory deterioration, expressed as an S/F ratio below 200 [10,11,12]. LUS evaluation was executed by an expert neonatologist, trained in LUS and with expertise in the LUS-guided recruitment maneuver, using a high-frequency linear probe (9–16 MHz) with General Electric Medical System LOGIQ S8 or with Mindray DC8-EXP ultrasound machines.

The control group (Oxygenation-guided group, Ox-group) was constituted by rescue LRMs conducted with the oxygenation-guided procedure, as operators trained in LUS and with expertise in the LUS-guided LRM were not available. This approach relies on the assumption that the recruitment of collapsed alveoli reduces the intrapulmonary shunt fraction and then improves oxygenation. In this procedure, oxygenation is adopted as an indirect bedside parameter for changes in lung volume.

The oxygenation-guided LRMs were performed during the same period and selected to match those of the LUSTR-group, in terms of type of pulmonary disease and severity of the disease (initial S/F ratio).

The placement of the saturation probe was preductal. The hemodynamic status was initially assessed through targeted neonatal echocardiography. Left ventricular dysfunction, pulmonary hypertension, or hypovolemia was evaluated and, eventually, treated before LRMs in both groups. In accordance with local guidelines, opioids (i.e., fentanyl, remifentanil) were administrated to the infants to provide comfort during mechanical ventilation. No muscle relaxant was given.

Infants enrolled were mechanically ventilated either with VN500 (Draeger, Lubecca, Germany) or with Sensor medics 3100A (Carefusion, San Diego, CA, USA).

This study was approved by the local ethics committee (CEROM, n. 4510 Prot. 8317/2021 I.5/16; 9 November 2021) and conducted in accordance with the Declaration of Helsinki.

### 2.2. Lung Recruitment Protocols

In the intervention group, LUS was systematically performed on all lung fields: anterior superior, lateral superior, posterior superior, anterior inferior, lateral inferior, and posterior inferior for the right and left sides [13]. If signs of lung consolidation or derecruitment were detected, after the exclusion of any modifiable causes of lung collapse (e.g., endotracheal tube misplacement), LUS-guided recruitment was initiated [7]. In accordance with previous reports in adults and children [1,14], the patient was positioned to maintain the most severely affected lung fields on the upper side, with constant ultrasound monitoring to visualize the progressive parenchymal reopening throughout the procedure in the most derecruited lung area. LRM was performed through the stepwise increase of airway pressure, either positive end-expiratory pressure (PEEP) during conventional mechanical ventilation (CMV) with volume guarantee or continuous distending pressure (CDP) during high-frequency oscillatory ventilation (HFOV). Airway pressure was augmented by 1 cmH_2_O every 1–2 min. The pressure was increased stepwise until the consolidation disappeared and was converted into a B2 or B1 pattern, and the corresponding level of airway pressure was defined as opening PEEP/CDP (Figure 1).

The procedure was planned to be interrupted in the case of clinical deterioration (i.e., increasing oxygen need, hemodynamic instability). PEEP/CDP was reduced stepwise until LUS began to show signs of lung consolidation or aeration loss in the lung field previously monitored. The corresponding airway pressure was defined as closing PEEP/CDP (Figure 1). The optimal PEEP/CDP was set right above the closing pressure. In the case of atelectasis development/persistence in other lung fields, the infant’s position was modified again to maintain the atelectatic lung field on the upper side and another LRM was repeated, again monitoring the collapsed area with ultrasound throughout the procedure. A comprehensive LUS follow-up was performed 1 h, 6 h, and 12 h after the end of the maneuver.

In the Oxygenation-guided group, the procedure in HFOV was performed as described in previous reports on neonates [6,15]. LRMs on conventional ventilation in neonates follow very different protocols [16]. Our internal protocol for LR while receiving conventional ventilation is derived from the one described in precedent reports in children and adults [17,18] and PEEP is increased by 1 cmH_2_O every 1–2 min. This protocol resulted in a similar pressure increase over time in both HFOV and CMV.

### 2.3. Data Collection

The following clinical data were retrospectively retrieved from medical records: birth weight (BW); gestational age (GA); clinical risk index for babies scoring system (CRIB-II score); sex; 5-min Apgar score; prenatal steroids administration; mode of delivery; chorioamnionitis; underlying lung disease, defined as (i) respiratory distress syndrome (RDS) [19], resistant to surfactant administration, (ii) neonatal acute respiratory distress syndrome (NARDS) [20], (iii) lung derecruitment during MV detected at chest X-ray or LUS; days of life at recruitment; incidence of bronchopulmonary dysplasia (BPD), defined in accordance with current guidelines [21]; length of stay; duration of mechanical ventilation; survival. The diagnosis of NARDS was made according to the Montreux definition [20] in the case of acute respiratory failure, without RDS or transient tachypnea of the newborn as primary conditions, with lung imaging suggestive of diffuse and bilateral lung edema not due to other causes (i.e., patent ductus arteriosus, pulmonary hemorrhage).

The following data were collected about LRMs: mode of ventilation; type of ventilator; FiO_2_, CDP/PEEP, SpO_2_, heart rate, and mean blood pressure at each time point (start, opening, and optimal pressure); duration of clinical/sonographic benefit, defined as hours without respiratory deterioration and/or echographic evidence of atelectasis; air leaks within 12 h from LRM; the need for chest drain for air leaks within 12 h from LRM. The presence of air leak was suggested by clinical deterioration, which prompted X-ray execution in the Ox-group; Whereas in the LUS-guided group, this was excluded by the follow-up LUS examinations. To assess the severity of the lung disease, we used the following parameters: the S/F ratio [10] and oxygen saturation index (OSI = CDP × FiO_2_/SpO_2_) [22].

In the LUSTR-group, the number of steps of pressure needed to obtain the S-pattern was collected. Ultrasound images and videos were retrospectively analyzed by two independent operators.

### 2.4. Study Bias

Selection bias was avoided including all the infants undergoing the rescue procedure (S/F) ratio below 200. Patients undergoing LUS-guided recruitment starting from a C-pattern and not showing signs of lung reopening (S-pattern development) were excluded from the study. To reduce the selection bias, the exclusion of the LUS-guided recruitment that did not develop the S-pattern was done in parallel by two authors (RC, EC) that were blinded to the outcome of the LRM. The authors demonstrated 100% agreement in the selection process.

The Ox-group was obtained by automatic matching for two different, relevant variables. There was no follow-up and therefore, no loss was experienced in this study. The multiple linear regressions were performed to correct for the most important confounding variables.

### 2.5. Primary and Secondary Outcomes

The primary outcome was the difference between the final and the initial S/F ratio (del-ta S/F ratio). A secondary efficacy outcome in patients undergoing HFOV ventilation was the difference between the final and the initial oxygen saturation index (delta OSI). A secondary safety outcome was the development of pneumothorax with and without the need for chest drain within 12 h from the LRM.

### 2.6. Statistical Analysis

To compare the characteristics across the groups, the Chi-square test of independence and the F-test were performed for categorical and continuous variables, respectively. A one-way nested ANOVA was conducted for each comparison group to determine whether the differences between the group means were statistically significant and to take into account the nested structure of the data as multiple observations are associated with the same patient. To this end, the response variable (Delta S/F) was log-transformed to normalize its distribution.

A multiple linear regression model was used to estimate the association between the S/F ratio and relevant explanatory variables. Initially, a model was built by including all the variables considered relevant to the impact of the procedure based on the theory (study groups, GA, starting S/F, starting CDP/PEEP, day of life, underlying lung disease, mode of ventilation, sex, change of position before LRM, CRIB-II score). Successively, we performed a stepwise selection process to obtain a reduced final model. The mean with the SD (standard deviation) and the median with the IQR (interquartile range) were reported for the characteristics related to the observations and to the patients, respectively. All model assumptions were checked and, in addition, the models were evaluated for multicollinearity, without detecting any relevant effect. As a result of the multiple linear regression, we reported the unstandardized regression coefficient (B) which describes how much the dependent variable is expected to increase when the related independent variable increases by one, holding all the other independent variables constant. We then reported the standard error for the unstandardized beta (SE B) in order to describe the variation around the estimates of the regression coefficient, the *t*-test statistic (t), and the *p*-value.

We performed an internal analysis of the LRs performed with the standard method and found a mean delta S/F ratio of 70 ± 30 during the procedure. Assuming an improvement of 50% with LUS guidance, the minimal sample size to detect significance was eight recruitments per group. However, to further avoid selection bias, we included all the LUS recruitment showing signs of lung recruitability during the study time. All the recruitments included in the LUSTR-group were matched through the software matching system with one control, based on the severity of the disease (initial S/F) and the type of lung disease. No missing data regarding the variable of interest (delta S/F) were found. Missing data regarding the independent variables were replaced with the median, whereas variables with a significant number of missing data were excluded from the analysis. A two-way repeated measures ANOVA was used to assess the difference in the oxygenation parameters and vital signs within the two groups across the timepoints. The offline inter-operator reliability was evaluated using the intraclass correlation coefficient (ICC). All statistical analyses were performed by a statistician (LA), using the statistical software R (Version 4.1.2, R Development Core Team, Vienna, Austria).

## 3. Results

Eighteen ultrasound-guided LRMs were performed during the study period (Figure 2A). Four of them were excluded, as there was no evidence of S-pattern development. No disagreements were reported in the offline assessment of the S-pattern (ICC = 1). The S-pattern developed within a median of two steps of pressure increase (IQR 1–3, range 1–4). Eighteen LRMs performed with the oxygenation-guided approach during the study period and meeting the inclusion criteria were found. Out of these, fourteen were best matched, as described in the methods section.

A total of thirteen patients were enrolled in the study (Figure 2B). Three of them underwent LRMs performed exclusively in accordance with the LUSTR protocol, while five received only oxygenation-guided LRMs. Five patients received both interventions, three of them initially undergoing ultrasound-guided LRM and the remaining initially undergoing the oxygenation-guided procedure. The main clinical characteristics of the patients enrolled are summarized in Table 1, while the procedural and clinical characteristics regarding the LRMs are reported in Table 2.

LRMs were more frequently carried out in patients undergoing HFOV (*n* = 26, 92.9%). The two LRMs (7.1%) during CMV were both performed using a patient-triggered pressure-control assist-control ventilation mode with volume guarantee; synchronization was provided by flow sensor. No signs of shunt fraction potentially impacting oxygenation deficit on the vascular side were detected in any of the cases at the time of the LRM. The univariate analysis (Table 2) and the multiple linear regression (Table 3) showed significantly improved respiratory outcomes in the LUSTR-group as compared to the Ox-group in terms of improvement of the S/F ratio (Delta S/F 110 ± 47.3 vs. 64.1 ± 52.6, crude *p* = 0.02, adjusted *p* = 0.009) and OSI (Delta OSI 8.2 ± 6.8 vs. 2.8 ± 2.2, *p* = 0.01).

The scatterplot graphs of the two outcomes are outlined in Figure 3A,B. No correlation was found between the underlying pulmonary condition and the initial respiratory picture (data not shown). The patient position was modified more frequently before the procedure in the LUSTR-group (64.3% vs. 7.1%, *p* = 0.02). The supine position was more common during the LRM in the Ox-group, while the lateral position was significantly more frequent in the LUSTR-group. Prone position was slightly more frequent in the LUSTR-group, although this difference was not statistically significant (Table 3).

The incidence of pneumothorax within 12 h from the end of the procedure as well as the need for chest drain (0 vs. 7.1%, *p* = 0.31) was lower in the LUSTR-group (0 vs. 14.2%, *p* = 0.15) (Table 2). These differences were not statistically significant. The outlier LRM in the Ox-group, showing a much higher delta S/F than the other ones (Figure 3A), was one of the two that led to the development of the pneumothorax in the subsequent 12 h. Interestingly, the opening pressure in the LUSTR-group was significantly higher than in the Ox-group (27.1 ± 2.7 cmH_2_O vs. 19.6 ± 4.9 cmH_2_O) (Table 2). The sonographic resolution of lung derecruitment was obtained in all the cases.

The hemodynamic condition was evaluated and optimized before the LRM. All the cases underwent an echographic evaluation that showed normalization of the hemodynamic conditions (i.e., resolution of the indirect signs of pulmonary hypertension, left ventricular dysfunction, or hypovolemia) before starting the LRM.

Neither group presented with respiratory or hemodynamic deterioration throughout the procedure, causing the suspension of the procedure or the need for additional interventions. Only the Ox-group presented a statistically significant, but clinically non-relevant, increase in the diastolic and mean blood pressure from the starting point (Figure 4). Since the systolic blood pressure did not change, the pulse pressure slightly decreased in this group, reaching the opening CDP/PEEP and returning to the starting value at the closing CDP/PEEP. No changes were detected in the heart rate, while both groups demonstrated a significant improvement in the oxygenation parameters throughout the procedure.

## 4. Discussion

Our study suggests that the LUS-guided recruitment displaying the S-pattern is associated with a higher success of the LUS-guided pulmonary recruitment in critically ill neonates suffering from severe respiratory insufficiency as compared to the standard oxygenation-guided procedure. The recruitments included in the LUSTR-group obtained delta S/F ratios with an average of 50 points higher than the LRM in the Ox-group (Table 3). Interestingly, the S-pattern appeared during the early phase of the procedure (with a maximum of four pressure steps, mostly after two pressure steps), making it a promising sonographic sign to guide the LRM.

A recent randomized controlled trial demonstrated the beneficial impact of LUS in guiding elective LRMs as opposed to the oxygenation-guided method, on a population of preterm infants before the first surfactant administration [6]. Infants enrolled in this trial were all suffering from RDS, a homogenous lung disease, known to respond well to lung recruitment, including oxygenation-guided maneuvers [15,23]. In fact, despite the fact that LUS-guided recruitment better mitigated lung inflammation and reduced the duration of mechanical ventilation, both groups showed a striking improvement in oxygen saturation and FiO_2_ after the LRM. In our study, LRMs were attempted as a rescue approach in the case of severe respiratory failure, demonstrated by the critically low starting S/F ratio as compared to the previous report [6]. The Ox-group did not show a very significant improvement in the S/F ratio and experienced a higher rate of air leaks. Patients with heterogeneous respiratory diseases, in terms of physiopathology and spatial distribution, may present a poor response to LRMs. According to our results, in this clinical setting LUS can consistently enhance the open lung ventilation strategy.

The incidence of air leak and the need for chest drain were lower, although not significantly (*p* = 0.15 and 0.31, respectively), in the LUSTR-group, despite reaching higher opening pressures. Interestingly, the starting CDP and the derived OSI were also higher in the LUSTR-group, suggesting a worse clinical condition and more aggressive initial ventilation setting. This result is consistent with previous reports comparing LUS-guided recruitment and oxygenation-guided recruitment in the animal model, in children, and in adults [1,2,14,24,25]. Hence, it could be argued that the higher opening PEEP/CDP may have itself been the cause of the respiratory amelioration in the LUSTR-group. In order to address this issue, we included the opening PEEP/CDP in the stepwise process of the linear regression, which ruled it out as a predictor of the success of the procedure. Moreover, updating the oxygenation-guided protocol to obtain higher opening pressures would not be feasible as no monitoring tools currently exist and may be detrimental in terms of the safety of the procedure [15,23].

The LRM in the Ox-group that showed a much higher delta S/F than the other ones (Figure 3b), was one of the two that developed the pneumothorax. This observation suggests a particularly severe lung disease responds well to the LRM, although this consequently makes it at a higher risk for pneumothorax.

The safety and the efficacy correlated with the LUS-guided recruitment may be due in part to the spatial detection of the collapsed parenchyma, allowing for LUS-guided postural recruitment. Indeed, the change in patient position before or during the procedure, to maintain the derecruited lung fields on the upper side, constituted a key feature of LUS-guided LRMs, which was performed in almost 80% of the cases, while it was episodical in the oxygenation-guided group (7%). In accordance with the gravity-dependent lung physiology [4,26], patient position guided by LUS allowed the opening pressure to be targeted towards the most derecruited parenchyma, improving the lung reopening of the collapsed areas and reducing the risk of overdistension of the aerated segments. Although no evidence is available to suggest that a particular body position during mechanical ventilation of the neonate is effective in producing sustained and clinically relevant improvement [27], it has been described in adult patients that lung recruitment performed in prone position may obtain better results than the supine position [28,29]. In our population, the prone position was more frequent in the LUSTR-group, although this difference was not significant; however, the difference in the lateral or supine position between the two groups was substantial, suggesting that the prone position itself could not justify the better results obtained in the LUSTR-group, while the LUS-guided postural recruitment is likely a key feature of the LUSTR protocol.

Performing lung ultrasounds to guide recruitment is an act of integrated point of care ultrasound (POCUS), as hemodynamics must be taken into account. In our study, some patients needed inotropic support of inhaled nitric oxide therapy. As suggested in previous reports [2], the hemodynamic derangements were corrected before starting the LRM [1,2].

The present study has limitations, including the retrospective design and the small sample size. Thoracic X-rays were not systematically performed before and after recruitment and overdistension could not be ruled out. In the LUSTR-group, the development of air leaks was excluded by LUS. In the Ox-group, the presence of air leaks was suggested by clinical deterioration, which prompted the X-ray execution. Less clinically relevant air leaks may have been not detected. Another limitation is the inclusion of different underlying diseases in a small population. Two LRMs (one per group) were performed in conventional ventilation. Therefore, the generalizability of the findings is limited to the least represented groups.

Including neonates with a starting S/F ratio below 200, even if previously reported in the literature as a threshold for high-risk respiratory conditions, was somehow arbitrary, although supported by previous studies [10,11,12]. Furthermore, it was not possible to correct the SpO_2_ value for the fetal hemoglobin content.

Patients undergoing LUS-guided recruitment and not showing signs of lung reopening were excluded from the study. On the contrary, no criteria are available to our knowledge to early interrupt the oxygen-guided recruitment protocols or to early detect the not-recruitable lung during oxygenation-guided recruitment. This is a limitation of our study as we could not infer how many patients have non-recruitable lungs in the Ox-group. Despite the increase in the selection bias, one of the strengths of the LUSTR protocol is its ability to target the intervention to the population showing signs of lung reopening. We believe that this is an important step toward the optimization of the procedure.

The primary outcome is focused on the respiratory improvement given by the specific LRM. Other clinically relevant outcomes could not be compared as five patients had both kinds of LRMs. Indeed, the different interventions were not conducted in two separate periods. Performing the LRMs with the two different approaches was based on the availability of operators trained in LUS and with expertise in the LUS-guided LRM.

## 5. Conclusions

Based on the present study, the LUSTR protocol may be superior to the oxygenated-guided procedure to optimize open lung ventilation in critically ill neonates. The potential to early detect lung recruitability and perform postural recruitment are the cornerstones of this novel lung recruitment protocol. This novel procedure may have the potential to reduce ventilator-induced lung injury and improve the overall outcome. Further large prospective trials are needed to confirm our results and verify if more clinically relevant outcomes are affected by the LUSTR protocol in this patient population

## Figures and Tables

**Figure 1 children-09-01035-f001:**
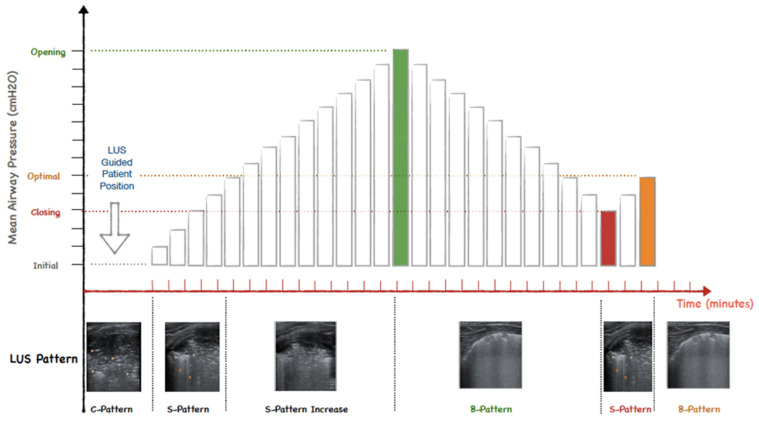
Lung UltraSound Targeted Recruited (LUSTR) protocol.

**Figure 2 children-09-01035-f002:**
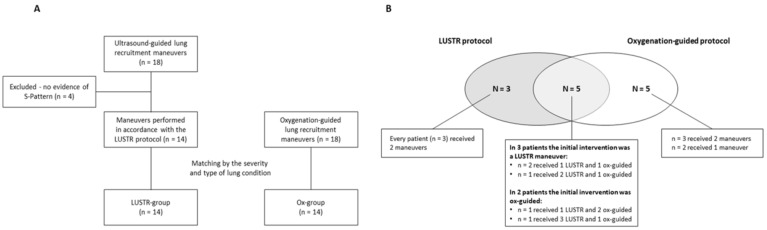
(**A**) Flow diagram for the study groups. (**B**) Venn diagram describing the distribution of the interventions among the study population.

**Figure 3 children-09-01035-f003:**
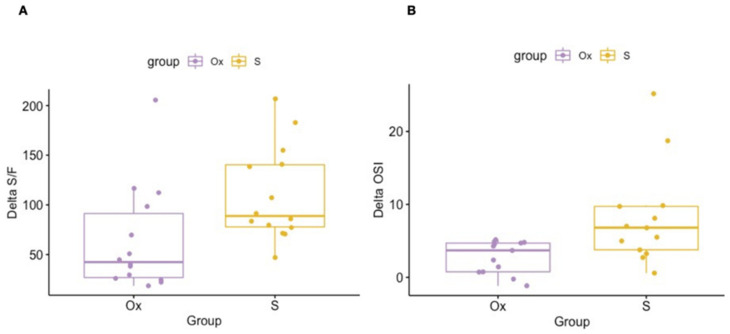
(**A**) difference between initial and final oxygen saturation/fraction of inspired oxygen ratio (Delta S/F) in the LUSTR-group and Ox-group. (**B**) difference between initial and final oxygen saturation index (Delta OSI) of the LUSTR-group and Ox-group.

**Figure 4 children-09-01035-f004:**
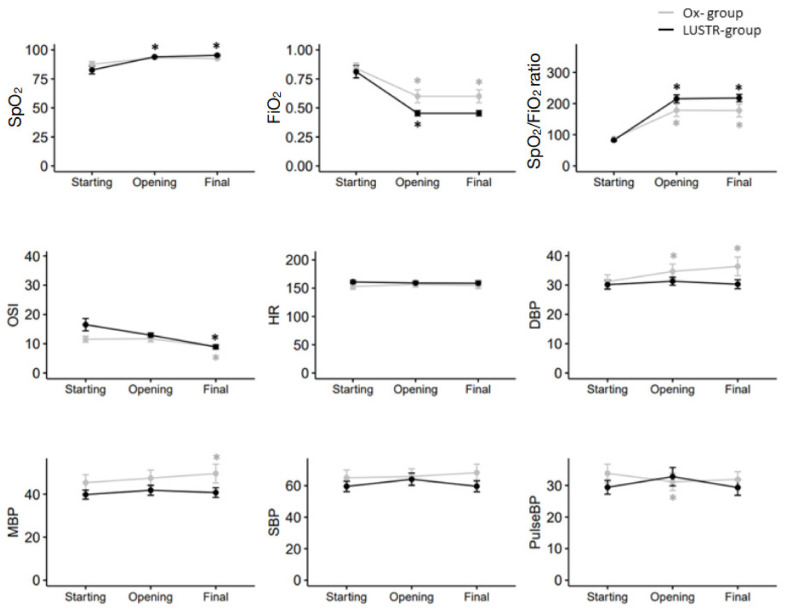
Trends of oxygen parameters and vital signs in the two groups throughout the procedure. The analysis to assess the variation of the different vital signs from the baseline (the starting point of the procedure) was performed by comparing them to the subsequent time points within the groups (opening vs. starting; final vs. starting). * Indicates significant (*p* < 0.05) variation from the starting parameter in the specific group.

**Table 1 children-09-01035-t001:** Clinical characteristics of the patients enrolled.

Characteristics	Study Population *n* = 13
Weeks of gestation, median [IQR]	26.5 (23.25–28)
BW in grams, median [IQR]	715 (552.5–1175)
Sex, n (%)	7 (53.8)
Prenatal steroids ^a^, n (%)	9 (75)
Vaginal delivery, n (%)	2 (15.4)
5-min Apgar score, median [IQR]	7 (6.5–8)
CRIB-II, median [IQR]	13 (11–15)
Chorioamnionitis, n (%)	5 (38.5)
Days of invasive ventilation, median [IQR]	18 (9–39.75)
Days of non-invasive ventilation, median [IQR]	63.5 (56.25–98)
BPD incidence ^b^, n (%)	7 (87.5)
Survival, n (%)	9 (69.3)
Days of hospital stay, median [IQR]	109.5 (99–181)
Day of life at recruitment, mean (SD)	14.4 (1.7)
SpO_2_ at recruitment, mean (SD)	85.1 (11)
FiO_2_ at recruitment, mean (SD)	0.83 (0.18)

IQR = interquartile range, GA = gestational age, BW = birth weight, CRIB-II = clinical risk index for babies, BPD = bronchopulmonary dysplasia. ^a^ The prenatal steroid administration refers to patients with GA below 37 (*n* = 12). ^b^ The BPD rate refers to patients with GA below 32 weeks only (*n* = 8).

**Table 2 children-09-01035-t002:** Characteristics of the LUSTR-group and Ox-group.

Characteristics	Ox-Group (*n* = 14)	LUSTR-Group (*n* = 14)	*p*
Day of life, mean (SD)	14.1 (10.5)	14.8 (7.9)	0.84
Lung condition, n (%)			0.90
Derecruitment during MV	5 (35.7)	5 (35.7)	
NARDS	5 (35.7)	5 (35.7)	
RDS	4 (28.6)	4 (28.6)	
Change in position, n (%)			
Before LRM	1 (7.1)	9 (64.3)	**0.02**
During LRM	0	2 (14.3)	0.46
Position during LRM, n (%)			
Supine	13 (92.9)	5 (35.7)	**0.001**
Prone	1 (7.1)	3 (21.4)	0.22
Lateral	0	5 (35.7)	**0.04**
Prone and lateral ^a^	0	1 (7.1)	0.31
SpO_2_, mean (SD)			
Starting	88 (9)	82 (12.5)	0.13
Final	92 (1.5)	95 (3)	**0.002**
FiO_2_, mean (SD)			
Starting	0.83 (0.18)	0.82 (0.19)	0.92
Final	0.6 (0.21)	0.45 (0.08)	**0.02**
S/F ratio, mean (SD)			
Starting	113.6 (36.5)	107.5 (38.1)	0.67
Final	177.7 (77.3)	217.5 (43.4)	0.1
Delta	64.1 (52.6)	110 (47.3)	**0.02**
OSI, mean (SD)			
Starting	11.6 (4)	16.9 (8.1)	**0.04**
Final	8.9 (3.1)	8.8 (6.8)	0.96
Delta	2.8 (2.2)	8.2 (6.8)	**0.01**
HFOV, n (%)	13 (92.9)	13 (92.9)	1
CDP (cmH_2_O), mean (SD)			
Starting	12.5 (3)	15.9 (4.5)	**0.03**
Opening	19.6 (4.9)	27.1 (2.7)	**<0.001**
Optimal	14.3 (2.5)	18.4 (4.2)	**0.007**
CMV, n (%)	1 (7.1)	1 (7.1)	1
PEEP (cmH_2_O), mean (SD)			
Starting	7	5	NA
Opening	9	10	NA
Optimal	6	7	NA
Duration of the procedure (min), median [IQR]	20 [15–31.2]	35 [30–40]	**0.002**
PNX, n (%)	2 (14.3)	0	0.15
Chest drain, n (%)	1 (7.1)	0	0.31

NARDS = neonatal acute respiratory distress syndrome; LRM = lung recruitment maneuver. SpO_2_ = oxygen saturation; FiO_2_ = fraction of inspired oxygen; S/F = SpO_2_/FiO_2_; OSI = oxygen saturation index; HFOV = high frequency oscillatory ventilation; CDP = continuous distending pressure; CMV = conventional mechanical ventilation; PEEP = positive end expiratory pressure; PNX = pneumothorax. ^a^ If the position was changed during the procedure from lateral to prone or vice versa. Statically significant results (*p* < 0.05) are reported in bold.

**Table 3 children-09-01035-t003:** Multiple linear regression for the log-transformed (delta) between the final and the initial oxygen saturation/fraction of inspired oxygen (SCR/F) ratio.

Variables	B	SE B	*t*	*p*
**LUSTR**	0.68	0.23	2.82	0.009
Underlying lung disease				
NARDS	0.62	0.24	2.56	0.017
RDS	0.38	0.34	1.13	0.27
Male sex	0.46	0.30	1.53	0.14
CRIB-II	0.09	0.06	1.168	0.10

B = the unstandardized beta; SE B = the standard error for the unstandardized. beta; *t* = the *t*-test statistic; *p* = the probability value; NARDS = neonatal acute respiratory distress syndrome; CRIB-II = clinical risk index for babies.

## Data Availability

The data that support the findings of this study are available on request from the corresponding author, M.P.

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
