# Peer review of "Lung UltraSound Targeted Recruitment (LUSTR): A Novel Protocol to Optimize Open Lung Ventilation in Critically Ill Neonates"

_children, 2022, doi:10.3390/children9071035_

Round 1

Reviewer 1 Report

This is a very well written paper I enjoyed reviewing. 

There are some minor text edits have to be done e.g. words split in the middle of a line. 

Author Response

Thank you for reviewing our paper and for your comment. We edited the typos as suggested.

Reviewer 2 Report

General comments:

The authors investigated the effectiveness of Lung UltraSound Targeted Recruitment (LUSTR) protocol on improving the success of lung recruitment maneuvers (LRMs). They found that the LUSTR protocol was associated with a higher success of the procedure in terms of Delta S/F as compared to the standard approach. This result remained significant after adjusting for confounding variables through multiple linear regressions. The authors conclude that LUSTR protocol may be a more effective and potentially safer option than the oxygenation-based procedure to guide open lung ventilation in neonates.

General concerns:

1.     Abbreviation: Please define abbreviations upon first appearance in the text and follow author’s instruction.

2.     Please correct many typos ‘mechani-cally’ etc. throughout the manuscript.

3.     Line36: Please correct ‘ventilator induced lung injury’ to ‘ventilator-induced lung injury’.

4.     Methods: Please define neonatal acute respiratory distress syndrome (NARDS).

5.     The OSI levels were significantly higher in non-survivors compared to survivors. Thus, the higher OSI, the worse prognosis. Please recheck the description of OSI in the Results, Table 2 and Discussion.

6.     Methods: Line 85: “All the infants were well-sedated during the maneuver”. Please describe the method of sedation.

7.     Figure S1: Please indicate the meaning of 0 and 1 in figure legend.

8.     In this study, the authors evaluated the short-term outcomes of LUSTR. Did the authors find any beneficial long-term effects of LUSTR?

Author Response

Thank you for the accurate review of our paper, and for the valuable comments that helped to improve the manuscript. About the concerns expressed by the reviewer:

  1. Abbreviation: Please define abbreviations upon first appearance in the text and follow author’s instruction.

As suggested, we defined the abbreviations more rigorously and consistently upon the first appearance in order to improve the clarity

  1. Please correct many typos ‘mechani-cally’ etc. throughout the manuscript.

We corrected the typos present in the manuscript

  1. Line36: Please correct ‘ventilator induced lung injury’ to ‘ventilator-induced lung injury’.

We modified the spelling as kindly suggested

  1. Methods: Please define neonatal acute respiratory distress syndrome (NARDS).

Thanks for this suggestion. We provided a clearer explanation of the criteria adopted for the diagnosis of NARDS.

  1. The OSI levels were significantly higher in non-survivors compared to survivors. Thus, the higher OSI, the worse prognosis. Please recheck the description of OSI in the Results, Table 2 and Discussion.

Thanks for this comment. We corrected the typo in Table 2 since, by mistake, we named the LUSTR-group as S-group. The LRMs performed in accordance with the LUSTR protocol had higher starting OSI levels, as the starting CDP was higher in this group, suggesting worse clinical conditions and the need for more “aggressive” ventilation. However, we could not compare the two strategies for outcomes, such as survival, as some patients received bot interventions.  We had mentioned this as a limitation of the study. We added the starting CDP in the linear regression through the stepwise process and did not show significance. We commented on this in the manuscript. We verified the description of OSI in the Results, Table 2 and Discussion.

  1. Methods: Line 85: “All the infants were well-sedated during the maneuver”. Please describe the method of sedation.

Thanks for this precise comment. In accordance with local guidelines, opioids are administrated to neonates receiving mechanical ventilation, to reduce pain and stress related to this invasive support. We commented on this in the manuscript, specifying the drugs used in our unit (fentanyl or remifentanil).

  1. Figure S1: Please indicate the meaning of 0 and 1 in figure legend.

We provided a more precise explanation in the figure legend, by changing 0 1 and 1 into Ox-group and LUSTR-group, respectively.

  1. In this study, the authors evaluated the short-term outcomes of LUSTR. Did the authors find any beneficial long-term effects of LUSTR?

Thanks for this thoughtful comment. Thirteen patients were enrolled in this study, receiving multiple separated LRMs. Since five of them received both kinds of interventions, we could not compare the clinical variables and long-term outcomes between the two groups. We had mentioned this as a limitation of the study and a potential perspective for further prospective studies